# Simple Synthesis of K_4_Nb_6_O_17_/C Nanosheets for High-Power Lithium-Ion Batteries with Good Stability

**DOI:** 10.3390/ma12020262

**Published:** 2019-01-15

**Authors:** Xiangwei Wang, Yunyun Zhai, Chunxia Kuang, Haiqing Liu, Lei Li

**Affiliations:** College of Biological, Chemical Sciences and Engineering, Jiaxing University, Jiaxing 314001, China; wangxiangwei@mail.zjxu.edu.cn (X.W.); zhaiyunyun@mail.zjxu.edu.cn (Y.Z.); kuangcx@mail.zjxu.edu.cn (C.K.)

**Keywords:** potassium niobate nanosheets, polydopamine, high rate performance, lithium-ion batteries

## Abstract

In this work, a series of two-dimensional (2D) large-size nanosheets were prepared through one-step exfoliation of the huge K_4_Nb_6_O_17_ crystals. The K_4_Nb_6_O_17_ nanosheets with the thickness of about 2 nm was used as the templates of dopamine polymerization and was then carbonized to form C-doped K_4_Nb_6_O_17_ nanosheets. More importantly, the C-doped K_4_Nb_6_O_17_ nanosheets exhibited excellent electrochemical performance with high specific capacity (381 mA h g^−1^ at 0.05 A g^−1^, 0.5–3.0 V vs. Li/Li^+^) and stable cyclability at high current density (remarkably, preserved a capacity of discharge approximately 90 mA h g^−1^ at 5 A g^−1^ after 1000 cycles). The good electrochemical performances of the C-doped K_4_Nb_6_O_17_ nanosheets can be attributed to the outstanding 2D structure and large specific surface, which afforded the short transport route for ion and electron. These noteworthy results demonstrated that the new 2D nanomaterials might be potential candidates for the high-performance, environmentally friendly, and low-cost electrochemical energy storage equipment.

## 1. Introduction

Advanced energy storage techniques [1,2] have drawn widespread attention since the need for matching the new electricity production technologies, such as solar, tidal, wind, etc [3]. Lithium-ion batteries (LIBs) have become a topic of international research due to their long cycling lifetime, high working voltage, high energy density, high rate capability, and environment-friendliness [4,5]. It could be an important complement for the insufficient of the other power sources, for instance, supercapacitors and fuel cells [6,7,8]. However, commercial graphite is hard to adapt to the demand of widely application duo to the slow Li^+^ diffusion rate and the formation of solid electrolyte interface (SEI) [9]. To solve the above-mentioned problem of commercial graphite anode, different electrode materials, for instance, multifarious carbon materials [10], mixed metal oxides [11], conducting polymers [12], et al. have been widely used as LIBs electrode materials. Unfortunately, the ion and electron transport paths of these materials are too long to achieve the desired results. Therefore, it is highly necessary to fabricate the novel electrode materials with high transmission efficiency.

The extensive research of graphene has evoked the attention to other two-dimensional (2D) materials. Particularly, metal oxide nanosheets are fascinating due to their wide variety of composition and structure. It is worth highlighting that niobium oxide, including NbO, NbO_2,_ Nb_2_O_3_, Nb_2_O_5_, and Nb_3_O_7_X (X = OH, F), have exhibited superior performance in many fields, such as catalysis, sensors, supercapacitors, solar cells, and LIBs [13,14,15]. T-Nb_2_O_5_/graphene composite papers showed a superior pseudocapacitor performance as free-standing electrodes and excellent rate capability [16]. A three-dimensional holey-graphene/niobia (Nb_2_O_5_) composite had an ultrahigh-rate energy storage at practical levels of mass loading (>10 mg cm^−2^) [17]. 

Among them, K_4_Nb_6_O_17_ has an ion-exchangeable layered structure, which own two kinds of alternating interlayer spaces (interlayers I and II), where K^+^ ions are located. The interlayers are hydrated in aqueous solution [18]. The cation-rich layered structure facilitates ion and electron transport, thus enhancing the transfer efficiency greatly. Altrathin SnNb_2_O_6_ nanosheets has been reported by a simple hydrothermal route with K_4_Nb_6_O_17_ nanosheets as the precursor [19]. Qinglin Deng [20] reported a hydrothermal reaction to acquire layered K_4_Nb_6_O_17_ anode material, which manifested a largish initial discharge specific capacity of approximately 513 mA h g^−1^ at 0.2 A g^−1^. However, as the number of cycle time increases, the capacity declines rapidly to 84 mA h g^−1^ because of the SEI film formation and structural collapse. Until now, few details about the application of K_4_Nb_6_O_17_ in LIBs were provided in previous research. To the best of our knowledge, how to improve the capacity of K_4_Nb_6_O_17_ at high current density has not been previously discussed.

In this work, we synthesized a range of 2D large-size K_4_Nb_6_O_17_ ultrathin-nanosheet, followed via the polymerization of carbon-rich dopamine precursor and ultimate carbonization of the K_4_Nb_6_O_17_-polydopamine composites. Due to the doped carbon, the conductivity of the K_4_Nb_6_O_17_/C nanosheets were improved, and thus enhanced the cyclability and rate capability. The cell based K_4_Nb_6_O_17_-C-800 nanosheets exhibited high initial capacity (381 mA h g^−1^ at 0.05 A g^−1^, with the voltage range of 0.5–3.0 V vs. Li/Li^+^). Moreover, the cell based K_4_Nb_6_O_17_-C-800 anode also possessed excellent rate performance (133 and 67 mA h g^−1^ at 1 and 5 A g^−1^, respectively) and stable cyclability (demonstrated a discharge capacity of approximately 150 mA h g^−1^ after 200 cycles, equivalent to approximately 84% of the 20th cycle at 0.1 A g^−1^), which were superior to the previously reported niobium oxide composites.

## 2. Materials and Methods

### 2.1. Materials

Nb_2_O_5_, K_2_CO_3_, *n*-Propylamine, dopamine (DA), *N*-methyl-2-pyrrolidone (NMP), poly-vinylidene fluoride (PVDF), and conductive carbon black were purchased from Sigma-Aldrich. A liquid electrolyte consisting of 1 M LiPF_6_ in ethylene carbonate/dimethyl carbonate/ethyl methyl carbonate (1/1/1, weight ratio) with the addition of 1 wt% vinylene varbonate was provided by Shenzhen Kejingstar Technology Ltd. (Shenzhen, China). Deionized water was used in all experiments, and all of the chemicals were used without being further purified.

### 2.2. Sample Preparation

The K_4_Nb_6_O_17_ single crystal was synthesized based on the flux method [21]. In a typical exfoliation process [22], K_4_Nb_6_O_17_ crystal (0.30 g), water (60 mL), and *n*-Propylamine (3.0 mL) was added into a Teflon-lined vessel and maintained at 120 °C for 3 d. The K_4_Nb_6_O_17_ ultrathin-sheets slurry obtained by low-speed centrifugation.

K_4_Nb_6_O_17_ ultrathin-sheet slurry (2 mL) was dissolved in deionized water (120 mL). Subsequently, a calculated amount of DA (the mass ratio of K_4_Nb_6_O_17_ and DA is 1:1) was dropped under stirring. After 1 h, NH_4_OH (0.3 mL, 28 wt%) was injected into the mixture and stirred for two days for the polymerization. The product was named as K_4_Nb_6_O_17_-PDA. It was rinsed with distilled water for at least three times. The composites were calcined in a horizontal resistance furnace under Ar at 800 °C for 2 h to obtain K_4_Nb_6_O_17_-C-800. With the same method, the K_4_Nb_6_O_17_-PDA was calcined at 400, 600, 700, 900, and 1000 °C, which were denoted as K_4_Nb_6_O_17_-C-400, K_4_Nb_6_O_17_-C-600, K_4_Nb_6_O_17_-C-700 K_4_Nb_6_O_17_-C-900, and K_4_Nb_6_O_17_-C-1000, respectively. For comparison, the K_4_Nb_6_O_17_ nanosheets were also annealed at 800 °C for 2 h and named as K_4_Nb_6_O_17_-800.

### 2.3. Characterization

The scanning electron microscopy (SEM, Hitachi S-4800, Tokyo, Japan) was used to measure the microstructure of all samples. The high-resolution transmission electron microscopy (HRTEM) was performed on a JEM-2100F (Tokyo, Japan). The phases of as-obtained samples were identified via X-ray diffractometer (XRD, DX-2600, Dandong, China) using Cu Kα radiation (λ = 0.15406 nm). Raman measurement was performed on an Invia-Reflrx Laser Micro-Raman spectrometer (Renishaw, London, England). The thickness of the K_4_Nb_6_O_17_ nanosheets was obtained via atomic force microscopy (AFM) images (Multimode Nanoscope IIIa, Tokyo, Japan).

### 2.4. Electrochemical Tests

To examine electrochemical properties of the respective samples, CR-2016 type cells (Kejingstar Technology Ltd, Shenzhen, China) were assembled with Li metal as the counter and reference electrodes simultaneously, and Celgard 2320 (Kejingstar Technology Ltd, Shenzhen, China) as the separator in a glovebox. A liquid electrolyte consisting of 1M LiPF6 in ethylene carbonate/dimethyl carbonate/ethyl methyl carbonate (1/1/1, *w*/*w*/*w*) with the addition of 1 wt% vinylene carbonate was provided by Shenzhen Kejingstar Technology Ltd. (Shenzhen, China). The working electrodes were prepared by blending the electroactive materials (70 wt%), conductive carbon black (20 wt%), and PVDF (10 wt%) in NMP. After that, the ultimate slurry was spread on the Cu foil and desiccated in vacuum at 60 °C for 24 h. In addition, the desiccated electrodes were roll-pressed. The cyclic voltammetry (CV) tests were measured at 0.1–1 mV s^−1^ within a range of 0.5–3.0 V using an electrochemical workstation (CHI 760E, CH Instruments Ins, Shanghai, China). The impedance test was carried out by an AC impedance analyzer (IM 6ex, Zahner, Kronach, Germany) with a frequency range of 1 M Hz-0.1 Hz. The as-made coin cells were cycled at 0.05–5 A g^−1^ using the Land CT2001A battery cycler (LAND, Wuhan, China) with the voltage range of 0.5–3.0 V.

## 3. Results

The preparation route was schematically described in Scheme 1. The synthesis of K_4_Nb_6_O_17_-C nanosheet consisted of exfoliated K_4_Nb_6_O_17_, the polymerization of the dopamine (DA), and carbonization. Firstly, the two-layer K_4_Nb_6_O_17_ ultrathin-sheets were acquired by a template-free and one-step synthesis. As we can see in Figure 1a, the sharp peaks at 9.5° and 22.2° are indexed to the (040) and (151) crystalline facets of K_4_Nb_6_O_17_ crystal, indicating that the degree of crystallization is high, and the spatial arrangement of the microchip layer is unconventional. After stripping by propylamine, the peaks that are located at 9.5° and 22.2° shift to 8.3° and 21.2°, respectively, and the diffraction peaks become broader and weaker. It is indicated that the layer of potassium citrate has been stripped by the insertion of propylamine molecules, and thus decreasing the crystal structure integrity and increasing the degree of disorder. The inset shows a big K_4_Nb_6_O_17_ crystal (2 mm) and the SEM of K_4_Nb_6_O_17_ crystal before exfoliated, there is an obvious thicker layered structure, which provides the possibility for further exfoliation to obtain the nanosheets. A few micron-sized debris during the treatment process also can be exfoliated. From Figure 1b, we can find that the zeta potential was between −20 mV to −58 mV at the pH from 2 to 12. It was very effective for the homogeneous adsorption and polymerization of DA on the surface of K_4_Nb_6_O_17_ nanosheets because of their negative charge (pH = 10, zeta potential = 47 mV). The inset of Figure 1b was K_4_Nb_6_O_17_ nanosheet dispersion. The dispersion was very stable after two weeks because of the electrostatic repulsion from abundant negative charges.

From Figure 2a, we can conclude ultrathin K_4_Nb_6_O_17_ nanosheets with lateral dimension that is about a dozen even several dozen microns can be observed, showing a typical 2D morphology. As indicated by the AFM image in Figure 2b, the thickness of as-prepared K_4_Nb_6_O_17_-C nanosheet was approximately 2 nm. Meanwhile, TEM characterization in Figure 2c,d further confirmed the characteristic of ultrathin thickness, which was beneficial for the transport of the ion and electron.

As we can see in Figure 3a–c, with the increase of calcination temperature, the thickness of K_4_Nb_6_O_17_-C slightly increased due to the formation of doped carbon and the overlay of slice layer in the thermal process. Whereas, the thickness had obviously turned to dozens of nanometers when the thermal temperature was higher than 800 °C (Figure 3d–e). From Figure 3f, we can find there was amorphous carbon onto the surface of crystallographic K_4_Nb_6_O_17_ nanosheets. The numbers of active bits and specific surface areas increased greatly due to the carbon coating, which will effectively improve the electrochemical performance. In Figure 4a, the two peaks at 1353 and 1598 cm^−1^ can be attributed to D- and G-bands of carbon, respectively. It is obvious that the pure K_4_Nb_6_O_17_ nanosheets do not have D- and G-bands. After carbonization, the intensity is converted from strong D-band to strong G-band as the carbonization temperature increases, which indicates that the degree of disorder decreases and the number of nanosheets increases with the carbonization temperature increase. The disorder degree decreases and recrystallization effect is obvious, the results correspond to XRD. It is reinforced that the evidence that the carbon with 2D architecture was converted from PDA after carbonization. The XRD diffraction patterns that are shown in Figure 4b illustrated that the crystals agree well with that of K_4_Nb_6_O_17_ of the JCPDS data file 21-1297. Also, the crystallinity of K_4_Nb_6_O_17_ was gradually increased with the increase of calcination temperature. However, after carbonizing at 400 °C and 600 °C, it was obvious that the PDA was coated on the surface and was incomplete carbonized, the uncarbonized PDA hindered the electron conduction. After carbonizing at 900 °C and 1000 °C, the nanosheets overlapped and the crystallinity was too high, which leads to the increase of diffusion channel of lithium ions and reduces the Li^+^ embedding and desorption efficiency during the charge-discharge process. After carbonizing at 800 °C, the PDA layer carbonized completely and the thickness of the K_4_Nb_6_O_17_-C nanosheet did not increase, which was beneficial for Li^+^/electron transport and the maintenance of the nanosheets structure. Moreover, HRTEM in Figure 3f shows that the crystallinity of K_4_Nb_6_O_17_ ultrathin-sheets was very high, which agreed with the XRD results. The coated amorphous carbon and flake structure increase the active bits of Li^+^/electron transport, and thus improving the cyclability and rate capability.

To investigate the electrochemical performances of lithium ions insertion/extraction, as-prepared K_4_Nb_6_O_17_-C-800 was measured using representative CV curves for the initial three cycles in the range of 0.5–3.0 V at 0.5 mV s^−1^. As we can see in Figure 5a, the oxidation peaks at about 1.00 and 1.75 V vs. Li^+^/Li were observed when the first oxidation process (Li^+^ extraction) and the reduction peak at approximately 1.21 V vs. Li^+^/Li was shown in the first reduction process (Li^+^ insertion). The following CV curves demonstrated quite good repeatability. As shown in Figure 5b, CV measurements at scan rates of 0.1 to 1 mV s^−1^ were used to explore the electrochemical characteristics of K_4_Nb_6_O_17_-C-800 electrode. There are a series of redox broad peaks, indicating that the Li^+^ insertion/extraction into/from K_4_Nb_6_O_17_-C-800 is a highly reversible process. The current peaks of cathodic and anodic shift noteworthily with the increase in scan speed, which may be attributed to the ohmic contributions of diffusion control process. The capacity of K_4_Nb_6_O_17_-C-800 is calculated via integrating the discharge process of CV, especially, the specific capacitance of K_4_Nb_6_O_17_-C-800 was 102 mA h g^−1^ at 1 mV s^−1^ (Figure 5c), showing a high-rate storage capability, which can be attributed to the high ionic conductivity and amorphous carbon coated layer structure of K_4_Nb_6_O_17_-C-800 nanosheets. Due to the special behavior of K_4_Nb_6_O_17_-C-800 nanosheets, the currents are in direct accordance with the scan rates in the CV tests, obeying the power law [23,24] ($i=kv^{b}$, where *k* and *b* both variable coefficients, *i* and *v* are current (A), and scan rate (mV s^−1^), respectively). The parameter *b* varies between 0.5 and 1, and the values of 0.5 and 1 are represented the diffusion-limited process (charge by Li^+^ insertion, battery-style) and the capacitive process (charge by surface capacitive effects, supercapacitor-style), respectively. As we can see in Figure 5d, the *b* value was 1 for the current peaks of cathodic and anodic at 0.1–1 mV s^−1^, manifesting a fast Li^+^ insertion process with a representative capacitive behavior. To differentiate quantitatively the effect of capacitance to the current response via the scan rate [25], we can use the formula, as follows:*i* = *k*_1_*v* + *k*_2_*v*^0.5^(1)
*i*/*v*^0.5^ = *k*_1_*v* + *k*_2_(2)
where $k_{1}$ and $k_{2}$ are both befitting values. The surface capacitive process manifests fast kinetics and can be represented as $k_{1}v$ (*b* = 1), while the diffusion-controlled process is represented as $k_{2}v^{0.5}$ (*b* = 0.5). By plotting $v^{0.5}$ versus $i/v^{0.5}$, $k_{1}$ and $k_{2}$ were determined from the slope and the Y-intercept. Based on this, the capacitive mechanisms with 0.1 mV s^−1^ contributed approximately 64% of the total capacity for K_4_Nb_6_O_17_-C-800 (Figure 5e). As shown in Figure 5f, the charge of surface effects increased sharply with the increase of sweep rate, from 64% at 0.1 mV s^−1^ (153 mA h g^−1^) to 95% at 1 mV s^−1^ (102 mA h g^−1^). This clearly indicated that the carbon doping increased the electron transport pathways, improving the electron transport property of the as-prepared K_4_Nb_6_O_17_-C-800 nanosheets, which was beneficial for the enhancement of the rate capability and cyclability.

Figure 6 showed the discharge-charge curves for the selected cycles at a current density of 0.1 and 1 A g^−1^ with a cut off potential window of 0.5–3.0 V, in which the voltage plateaus are in good agreement with the peaks of the CV curves in Figure 5a, even after extralong cycling. These outstanding electrochemical properties of K_4_Nb_6_O_17_-C-800 were further emphasized by rate capability tests in Figure 7a, the specific capacitance of K_4_Nb_6_O_17_-C-800 in the first discharge process obtain 382 mA h g^−1^. With the increase of current densities from 0.05 to 5 A g^−1^, the charge/discharge capacities of all the cells declined because of cell polarization. The drastic drop in capacity is often associated with inevitable irreversible loss of lithium ions for the formation of the solid electrolyte interface layer (SEI). Unlike in many other battery chemistries, such as the LiFePO_4_, where extra lithium is provided for in the electrode to compensate for amount of lithium used for the SEI layer formation, the K_4_Nb_6_O_17_-C electrode do not have that extra Li+ for this purpose [20,26,27,28,29]. The possible mechanisms of lithium ion intercalation:(3)$$Nb^{5+}+e^{-}\to Nb^{4+}$$
(4)$$K_{4}Nb_{6}O_{17}+Li^{+}+e^{-}\to LiK_{4}Nb_{6}O_{17}$$
(5)$$C+xLi^{+}+xe^{-}\to Li_{x}C$$

The cell using K_4_Nb_6_O_17_-C-800 nanosheets displayed the highest discharge capacities at all current densities, after undergoing a high current density of 5 A g^−1^ (67 mA h g^−1^), a special capacity of 186 mA h g^−1^ can be sustained, while the current density returns to 0.1 A g^−1^ again, which indicates excellent reversibility and cycling stability of the K_4_Nb_6_O_17_-C-800 nanosheets. As shown in Figure 7b, K_4_Nb_6_O_17_-C-800 exhibited high reversible capacities of 102 mA h g^−1^ and 85 mA h g^−1^ undergoing 1000 cycles at 1 A g^−1^ and 5 A g^−1^, respectively. The improvement can be attributed to the increase of conductivity and electron transfer property due to the introduction of carbon coating.

To understand the cycle retention characteristics, Figure 7c gives the cycling test that was conducted at 0.1 A g^−1^. Despite that some capacity decreased after 200 cycles, the K_4_Nb_6_O_17_-C-800 electrode still preserved a discharge capacity of approximate 150 mA h g^−1^, correspond approximately 84% of the 20th cycle (201 mA h g^−1^), and the coulombic efficiency was excellent (100.6%). It should be emphasized that the specific capacity of K_4_Nb_6_O_17_-C-800 was also superior to those of many reported Nb_2_O_5_ nanomaterials (see comparison in Table 1). Semicircle areas in the Nyquist plots at different frequencies refer to different aspects of electrochemical. Semicircle at high frequency represents the contact resistance, and that at medium frequency represents the charge transfer resistance. The straight line at low frequency represents the mass transfer of Li-ions [26]. As shown in Figure 7d, obviously, the diameter of the semicircle for K_4_Nb_6_O_17_-C-800 was the smallest when compared with other samples, which demonstrates that K_4_Nb_6_O_17_-C-800 owns the lower contact and charge transfer resistances, due to the increased conductivity of carbon doping. In the other hand, obviously, the diameter of the semicircle increases after 200 cycles with a current density of 0.1 A g^−1^, as compared with fresh uncycled cells, which demonstrates that K_4_Nb_6_O_17_-C possesses the lower contact and charge transfer resistances at the original state. The increase in resistance at the electrode/electrolyte interface for K_4_Nb_6_O_17_-C anode affected the lithium ion kinetics, which is correlated with the steady loss in the cell capacity during the initial cycles and continued to the 200th cycle. Due to the outstanding 2D structure and large specific surface, which afforded the short transport route for ion and electron and provided a large number of active sites for Li ion storage and shorter lithium ion transfer distance. All of these attributes contributed immensely to the improved cycling and performance of the K_4_Nb_6_O_17_-C composite electrode produced.

## 4. Conclusions

We have reported a facile design and synthesis of the strongly coupled K_4_Nb_6_O_17_-C nanosheets by one-step exfoliating and dopamine polymerization for carbonizing. Benefiting from the unique structural characteristic, the synthesizing hybrid nanosheets exhibited excellent performance in LIBs. Moreover, the electrode material showed pronounced capacities and stability at high current density. The facile procedure for the synthesis of the K_4_Nb_6_O_17_-C nanosheets, together with its prominent electrochemical performance, may motivate the exploration of more 2D nanostructured carbon doped metal oxide hybrid materials for energy storage related applications.

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
