# Peer review of "Simple Synthesis of K4Nb6O17/C Nanosheets for High-Power Lithium-Ion Batteries with Good Stability"

_materials, 2019, doi:10.3390/ma12020262_

Round 1

Reviewer 1 Report

Manuscript Number: materials-400385 by Wang et al. titled "Facile synthesis of K4Nb6O17/C nanosheets for high-2 power lithium-ion batteries with superstability"

This is a manuscript based on 33 references. The results are interesting. This reviewer is happy to recommend this manuscript to be published in Materials after a major revision. Below are the reviewer’s comments.

Please revise and proofread the title and the abstract of the manuscript by an English language expert. [superstability?]? [Facile?].

The authors need also to revise and proofread the abstract [The new electrode materials with high specific capacity and stable cyclability at high 9 current are important for the development of lithium-ion batteries (LIBs). Please move this sentence to the introduction?? which afforded the short transport route??] and [environmentally friendly and low-cost electrochemical energy storage equipment??].

The authors mentioned that the cut-off voltage was 3.0 V. However, it is observed in Figures 6a and 6b that the voltage at the first discharge cycle is low…almost 1.7 V at 0.1 A/g and around 2 V at 1 A/g…why? Why the potential is low at the first cycle and higher at the 2nd cycle and beyond (figure 4a??)

What was the contribution of carbon on the overall capacity of the K4Nb6O17-C anode? Fig 7b shows that the K4Nb6O17-800 delivers less than 50 mAh/g after 200 cycles while the K4Nb6O17-C-800 anode delivers higher capacity…why? What is K4Nb6O17 in Figure 7a?

The authors mentioned that the K4Nb6O17-C anode exhibited excellent performance in LIBs. The proposed anode delivered lower capacity than graphite anode and even it can compete with commercial anodes in LIBs. The authors need to elaborate more on this in the introduction and conclusion sections. Note that the working potential of the K4Nb6O17/C anode is high, hence it is not beneficial when used as anode in Li-ion full cells.

Please use the same colors for the curves shown in Figure7 (a, b and d). For example, the red curve in figure 7b is for K4Nb6O17-C-800 while the red curve in Fig 6a is for K4Nb6O17-C-600.

The CV peaks in Fig 5 do not correspond to the voltage plateaus in Fig 6. Why? The authors need to provide reaction mechanism of Li with K4Nb6O17-C anode and discuss it in the revised manuscript (redox or electrochemical reactions).

Transition metal oxides based on Li-alloy reaction mechanism exhibit a high irreversible capacity at the first cycle (see: Electrochimica Acta, 224, PP. 608-621, (2017), Adv. Energy Mater. 2016, 1502159 and Journal of Alloys and Compounds, 2016, 686, 733; Front Chem. 2018; 6: 166 (doi:  10.3389/fchem.2018.00166) and RSC Adv., 2018, 8, 5189-5196, Solid State Ionics, 286 (2016) 72–82. Fig 6 (a and b) shows a high loss in the capacities at the first cycle. The authors need to discuss the cause of loss in the discharge capacity (see/cite the refs above) . What is the effect of the high surface area and thickness of K4Nb6O17anode on the capacity at the first cycle?  The coulombic efficiency in Figure 7b does not reflect the loss in capacity at the first cycle why? There is a loss of about 40%-50% in capacity at the first cycle. This needs to be discussed in the revised manuscript.

The authors need to report results on the electrochemical impedance (Nyquist plots), of the K4Nb6O17-C anode before and after cycling.

This manuscript contains many English errors and typos and should be revised and proofread by an English language expert.  For example, synthesis of the strongly coupled K4Nb6O17-C nanosheets by one-step exfoliating and dopamine polymerization for carbonizing? the synthesizing hybrid nanosheets exhibited?

Author Response

Comments and Suggestions for Authors

Manuscript Number: materials-400385 by Wang et al. titled "Facile synthesis of K4Nb6O17/C nanosheets for high-power lithium-ion batteries with superstability"

This is a manuscript based on 33 references. The results are interesting. This reviewer is happy to recommend this manuscript to be published in Materials after a major revision. Below are the reviewer’s comments.

(1)Please revise and proofread the title and the abstract of the manuscript by an English language expert. [superstability?]? [Facile?].

Response:

Thanks for the valuable comment. According to your kindly suggestions, we have revised our title and abstract of the manuscript.

(2)The authors need also to revise and proofread the abstract [The new electrode materials with high specific capacity and stable cyclability at high current are important for the development of lithium-ion batteries (LIBs). Please move this sentence to the introduction?? which afforded the short transport route??] and [environmentally friendly and low-cost electrochemical energy storage equipment??].

Response:

Thanks for the valuable comment. We have revised our abstract and deleted this sentence. Also 2D C-doped K4Nb6O17 nanosheets afforded the short transport route for ions and electrons. The noteworthy results demonstrated the new 2D nanomaterials may be potential candidate for the high-performance, environmentally friendly and low-cost electrochemical energy storage equipment, such as lithiums ion batteries, supercapacitors, electrocatalysis, et al.

(3)The authors mentioned that the cut-off voltage was 3.0 V. However, it is observed in Figures 6a and 6b that the voltage at the first discharge cycle is low…almost 1.7 V at 0.1 A/g and around 2 V at 1 A/g…why? Why the potential is low at the first cycle and higher at the 2nd cycle and beyond (figure 4a??)

Response:

The first discharge voltage is the open circuit voltage of battery, which should be related to the battery stasis process. Moreover, because of the electrode is manually coated, it is difficult to guarantee the equal quality of the active material. So the first discharge voltage between different batteries is inconsistent.

When the second discharge cycle, the charge and discharge capacity and voltage are equivalent, therefore, it is higher than the first discharge voltage.

(4)What was the contribution of carbon on the overall capacity of the K4Nb6O17-C anode? Fig 7b shows that the K4Nb6O17-800 delivers less than 50 mAh/g after 200 cycles while the K4Nb6O17-C-800 anode delivers higher capacity…why? What is K4Nb6O17 in Figure 7a?

Response:

There are two functions of carbon. First, the structure of K4Nb6O17 is well protected during the high temperature treatment due to the coating of carbon source PDA. The sheet structure remains better and enhances the specific surface area. Second, the conductivity of the negative electrode material is further improved due to the carbon loading. So the role of carbon is to protect the two-dimensional structure and increase the conductivity.

  The K4Nb6O17, without loaded with carbon, has poor performance due to damages of the flake structure at high temperature. From the Figure 4b, we can find the crystallization degree of K4Nb6O17-800 is improved compared with K4Nb6O17-C-800. So the 2D nanostructure is destroyed. The lithium ions cannot easily inset into the K4Nb6O17-800 crystal. The capacity of K4Nb6O17-800 is lower than K4Nb6O17-C-800.

The K4Nb6O17 in Figure 7a was pure K4Nb6O17 nanosheets without any embellishment.

(5)The authors mentioned that the K4Nb6O17-C anode exhibited excellent performance in LIBs. The proposed anode delivered lower capacity than graphite anode and even it can compete with commercial anodes in LIBs. The authors need to elaborate more on this in the introduction and conclusion sections. Note that the working potential of the K4Nb6O17/C anode is high, hence it is not beneficial when used as anode in Li-ion full cells.

Response:

 Thanks for the valuable comment. The capacity of K4Nb6O17-C anode is much higher than commercial graphite anode (about 80-100 mAh g-1 at 0.1 A g-1). Especially, at the high current densities, the capacity of K4Nb6O17-C anode is valuable. Although the working potential of the K4Nb6O17-C anode is high, the capacities of K4Nb6O17-C anode at high current densities and stability are worthy for us to probe.

(6)Please use the same colors for the curves shown in Figure7 (a, b and d). For example, the red curve in figure 7b is for K4Nb6O17-C-800 while the red curve in Fig 6a is for K4Nb6O17-C-600.

Response:

 Thanks for the valuable suggestions. The color of the K4Nb6O17-C at the same temperature has been unified. For example, the dark cyan curve means K4Nb6O17-C-800 in Fig.4a, Fig.4b and Fig.7a, Fig.7b, Fig.7d.

(7)The CV peaks in Fig 5 do not correspond to the voltage plateaus in Fig 6. Why? The authors need to provide reaction mechanism of Li with K4Nb6O17-C anode and discuss it in the revised manuscript (redox or electrochemical reactions).

Response:

  As we can see in Figure 5a, the oxidation peaks at about 1.00 and 1.75 V vs. Li+/Li were observed when the first oxidation process (Li+ extraction), and the reduction peak at approximate 1.21 V vs. Li+/Li was showed in the first reduction process (Li+ insertion). As shown in figure 6a, the voltage plateaus at about 0.95 V and 1.7 V in the charge and 1.23 V in the discharge process. Because the CV peak shape is relatively flat, the voltage plateaus are not prominent.

  The reaction mechanism of Li with K4Nb6O17-C anode was discussed in the revised manuscript  (in the line 232 to 235 of page 7).

The possible mechanisms of lithium ion intercalation:

(8)Transition metal oxides based on Li-alloy reaction mechanism exhibit a high irreversible capacity at the first cycle (see: Electrochimica Acta, 224, PP. 608-621, (2017), Adv. Energy Mater. 2016, 1502159 and Journal of Alloys and Compounds, 2016, 686, 733; Front Chem. 2018; 6: 166 (doi:10.3389/fchem.2018.00166) and RSC Adv., 2018, 8, 5189-5196, Solid State Ionics, 286 (2016) 72–82. Fig 6 (a and b) shows a high loss in the capacities at the first cycle. The authors need to discuss the cause of loss in the discharge capacity (see/cite the refs above). What is the effect of the high surface area and thickness of K4Nb6O17anode on the capacity at the first cycle?  The coulombic efficiency in Figure 7b does not reflect the loss in capacity at the first cycle why? There is a loss of about 40%-50% in capacity at the first cycle. This needs to be discussed in the revised manuscript.

Response:

The drastic drop in capacity is often associated with inevitable irreversible loss of lithium ions for the formation of the solid electrolyte interface layer (SEI). Unlike in many other battery chemicals such as the LiFePO4, where extra lithium is provided for in the electrode to compensate for amount of lithium used for the SEI layer formation, the K4Nb6O17-C electrode do not have that extra Li+ for this purpose.

(Solid State Ionics, 286 (2016) 72–82; Electrochimica Acta, 224, PP. 608-621, (2017); RSC Adv., 2018, 8, 5189-5196; small 2017, 1603610)

(9)The authors need to report results on the electrochemical impedance (Nyquist plots), of the K4Nb6O17-C anode before and after cycling.

Response:

  Thank you very much for your advice. We have report the results on the electrochemical impedance (Nyquist plots) of the K4Nb6O17-C anode before and after cycling. As shown in Figure 7d, obviously, the diameter of the semicircle for K4Nb6O17-C-800 was the smallest compared with other samples, which demonstrates that K4Nb6O17-C-800 owns the lower contact and charge transfer resistances, duo to the increased conductivity of carbon doping. In the other hand, obviously, the diameter of the semicircle increases about from 110 Ω to 140 Ω after 200 cycles with a current density of 0.1 A g-1, compared with fresh uncycled cells.

(10)This manuscript contains many English errors and typos and should be revised and proofread by an English language expert.  For example, synthesis of the strongly coupled K4Nb6O17-C nanosheets by one-step exfoliating and dopamine polymerization for carbonizing? The synthesizing hybrid nanosheets exhibited?

Response:

  Thank you very much for your advice. We have advised our manuscript by an English language expert.

Reviewer 2 Report

This manuscript describes to synthesize of K4Nb6O17/C 2D large-size nanosheets prepared by one-step exfoliation of the K4Nb6O17 crystal. The synthesis attempt for the 2D nanomaterial anode active materials has a high performance for electrochemical performance as energy storage materials. The authors express that an anode electrode with K4Nb6O17/C composites can obtain a charge-discharge capacity of approximately 381 mAh/g at 0.05 A/g and 90 mAh/g at 5 A/g, which overcomes the already reported capacities of another Nb-oxidation anode. However, the authors’ original scope of the synthesis of K4Nb6O17/C composite in this study is interest as engineering results not so remarkable scientific study, so the authors should mainly clarify the scientific locations of the effort among the authors’ results. For above reasons, this paper needs some minor revisions for the publication standard of this journal at the moment.

1.       K4Nb6O17 nanosheet had no peak in Fig.1(a), however, the K4Nb6O17-800 (K4Nb6O17 after 800℃ anneal treatment) had some peaks in XRD of Fig.4(b). Is it re-crystallization? The optimized anode material shown in your article was really nanosheet?

2.       How about are the Raman spectra of K4Nb6O17/C samples in another anneal conditions? The reviewer could find that the crystallinity of K4Nb6O17-C-900 and 1000 were same as K4Nb6O17-800 or K4Nb6O17-C-800 from the results of XRD in Fig.4(b). The authors should show the carbonization of K4Nb6O17-C-400, 600, 900, and 1000.

3.       Please show the specifications of electrolyte used in your test cell and the charge-discharge cycle measurement conditions in your cycle stability results, CCCV or CC, SOC, DOD etc.

4.       The authors express as the follows in the line 234 to 237 of page 8, “ As shown in Figure 7(d), K4Nb6O17-C-800 exhibits high reversible capacities…”. This sentence is likely to explain the cycle properties, however, Fig.7(d) is only Nyquist plots. Please show your scientific or engineer scope for the Nyquist plot results of Fig.7(d). 

Author Response

Comments and Suggestions for Authors

This manuscript describes to synthesize of K4Nb6O17/C 2D large-size nanosheets prepared by one-step exfoliation of the K4Nb6O17 crystal. The synthesis attempt for the 2D nanomaterial anode active materials has a high performance for electrochemical performance as energy storage materials. The authors express that an anode electrode with K4Nb6O17/C composites can obtain a charge-discharge capacity of approximately 381 mAh/g at 0.05 A/g and 90 mAh/g at 5 A/g, which overcomes the already reported capacities of another Nb-oxidation anode. However, the authors’ original scope of the synthesis of K4Nb6O17/C composite in this study is interest as engineering results not so remarkable scientific study, so the authors should mainly clarify the scientific locations of the effort among the authors’ results. For above reasons, this paper needs some minor revisions for the publication standard of this journal at the moment.

(1)K4Nb6O17 nanosheet had no peak in Fig.1(a), however, the K4Nb6O17-800 (K4Nb6O17 after 800℃ anneal treatment) had some peaks in XRD of Fig.4(b). Is it re-crystallization? The optimized anode material shown in your article was really nanosheet?

Response:

As a result of high temperature treatment, as shown in figure (2a, 3c), the 2D nanostructure of K4Nb6O17 will be destroyed. With the gradual increase of temperature, the stack becomes more and more serious and recrystallization, resulting in diffraction peaks in the XRD. Therefore, we used PDA wrapping to prevent this phenomenon. As can be seen from figure (3a-3e), although recrystallization occurs, the 2D morphology is still maintained and the lamellar layer is increased.

(2)How about are the Raman spectra of K4Nb6O17/C samples in another anneal conditions? The reviewer could find that the crystallinity of K4Nb6O17-C-900 and 1000 were same as K4Nb6O17-800 or K4Nb6O17-C-800 from the results of XRD in Fig. 4(b). The authors should show the carbonization of K4Nb6O17-C-400, 600, 900, and 1000.

Response:

  Thank you very much for your advice, we have report the results on the Raman spectra of K4Nb6O17/C samples in another anneal conditions. As shown in Figure 4a, it is obvious that the pure K4Nb6O17 nanosheets do not have D- and G- bands, after carbonization, the intensity is converted from strong D-band to strong G-band as the carbonization temperature increases, which indicates that the degree of disorder decreases and the number of nanosheets increases with the carbonization temperature increases. The disorder degree decreases and recrystallization effect is obvious, the results correspond to XRD. It is reinforced that the evidence that the carbon with 2D architecture was converted from PDA after carbonization.

(3)Please show the specifications of electrolyte used in your test cell and the charge-discharge cycle measurement conditions in your cycle stability results, CCCV or CC, SOC, DOD etc.

Response:

The Celgard 2320 membrane was used. A liquid electrolyte consisting of 1M LiPF6 in ethylene carbonate/dimethyl carbonate/ethyl methyl carbonate (1/1/1, w/w/w) with the addition of 1 wt% vinylene carbonate was provided by Shenzhen Kejingstar Technology Ltd.

(4)The authors express as the follows in the line 234 to 237 of page 8, “ As shown in Figure 7(d), K4Nb6O17-C-800 exhibits high reversible capacities…”. This sentence is likely to explain the cycle properties, however, Fig.7(d) is only Nyquist plots. Please show your scientific or engineer scope for the Nyquist plot results of Fig.7(d).

Response:

Several statements that we made were more ambiguous than intended, and we have adjusted the text to be clearer.

  Semicircle areas in the Nyquist plots at different frequencies refer to different aspects of electrochemistry. Semicircle at high frequency is the contact resistance by interfacial films, that at medium frequency is the charge transfer resistance, and the straight line at low frequency is the mass transfer of Li-ions. As shown Figure7d, the diameter of the semicircle for K4Nb6O17-C-800 was the smallest compared with other samples, which demonstrates that K4Nb6O17-C-800 owns the lower contact and charge transfer resistances, duo to the increased conductivity of carbon doping.

Reviewer 3 Report

The article "Facile synthesis of K4Nb6O17/C nanosheets for high-power lithium-ion batteries with superstability" by X. Wang et al described the preparation of C-doped K4Nb6O17 nanosheets exhibited excellent electrochemical performance with high specific capacity and stable cyclability. I would recommend it for acceptance after the considering the listed below.

1. In Figure 2(b), the thickness of prepared K4Nb6O17-C nanosheet was determined approximately 2 nm determined by the AFM image. However, the profile is unclear. Please show the more clear and large size profile for determine the thickness. It seems like the thickness is 4 nm.

2. In the experimental section, the procedure of calcination at 400, 600 and 1000°C were not described. And, please explain the results of the procedure of calcination at 700°C (it is only described at the capacity results). Please obtain the consistency between the experimental and results.

Author Response

The article "Facile synthesis of K4Nb6O17/C nanosheets for high-power lithium-ion batteries with superstability" by X. Wang et al described the preparation of C-doped K4Nb6O17 nanosheets exhibited excellent electrochemical performance with high specific capacity and stable cyclability. I would recommend it for acceptance after the considering the listed below. (1)In Figure 2(b), the thickness of prepared K4Nb6O17-C nanosheet was determined approximately 2 nm determined by the AFM image. However, the profile is unclear. Please show the more clear and large size profile for determine the thickness. It seems like the thickness is 4 nm. Response: The more clear and large size profile for determine the thickness, it can be seen from the machine display results, the thickness of K4Nb6O17 nanosheet was 2.09 nm. (2)In the experimental section, the procedure of calcination at 400, 600 and 1000°C were not described. And, please explain the results of the procedure of calcination at 700°C (it is only described at the capacity results). Please obtain the consistency between the experimental and results. Response: The treatment was same with 800°C, through the horizontal tubular furnace at 700°C and preservated 2 h.

Round 2

Reviewer 1 Report

The reviewer is satisfied with the corrections and answers from the authors and agrees to publish this paper in Materials

Reviewer 3 Report

The author answered my question well, and the article
could be published in "materials".

Best wishes!

Happy new year!